# Softmax Q-Distribution Estimation for Structured Prediction: A Theoretical Interpretation for RAML

## Abstract

Reward augmented maximum likelihood (RAML), a simple and effective learning framework to directly optimize towards the reward function in structured prediction tasks, has led to a number of impressive empirical successes. RAML incorporates task-specific reward by performing maximum-likelihood updates on candidate outputs sampled according to an exponentiated payoff distribution, which gives higher probabilities to candidates that are close to the reference output. While RAML is notable for its simplicity, efficiency, and its impressive empirical successes, the theoretical properties of RAML, especially the behavior of the exponentiated payoff distribution, has not been examined thoroughly. In this work, we introduce softmax Q-distribution estimation, a novel theoretical interpretation of RAML, which reveals the relation between RAML and Bayesian decision theory. The softmax Q-distribution can be regarded as a smooth approximation of the Bayes decision boundary, and the Bayes decision rule is achieved by decoding with this Q-distribution. We further show that RAML is equivalent to approximately estimating the softmax Q-distribution, with the temperature $\tau$ controlling approximation error. We perform two experiments, one on synthetic data of multi-class classification and one on real data of image captioning, to demonstrate the relationship between RAML and the proposed softmax Q-distribution estimation method, verifying our theoretical analysis. Additional experiments on three structured prediction tasks with rewards defined on sequential (named entity recognition), tree-based (dependency parsing) and irregular (machine translation) structures show notable improvements over maximum likelihood baselines.

## 1 Introduction

Many problems in machine learning involve structured prediction, i.e., predicting a group of outputs that depend on each other. Recent advances in sequence labeling (Ma & Hovy, 2016), syntactic parsing (McDonald et al., 2005) and machine translation (Bahdanau et al., 2015) benefit from the development of more sophisticated discriminative models for structured outputs, such as the seminal work on conditional random fields (CRFs) (Lafferty et al., 2001) and large margin methods (Taskar et al., 2004), demonstrating the importance of the joint predictions across multiple output components.

A principal problem in structured prediction is direct optimization towards the task-specific metrics (i.e., rewards) used in evaluation, such as token-level accuracy for sequence labeling or BLEU score for machine translation. In contrast to maximum likelihood (ML) estimation which uses likelihood to serve as a reasonable surrogate for the task-specific metric, a number of techniques (Taskar et al., 2004; Gimpel & Smith, 2010; Volkovs et al., 2011; Shen et al., 2016) have emerged to incorporate task-specific rewards in optimization. Among these methods, reward augmented maximum likelihood (RAML) (Norouzi et al., 2016) has stood out for its simplicity and effectiveness, leading to state-of-the-art performance on several structured prediction tasks, such as machine translation (Wu et al., 2016) and image captioning (Liu et al., 2016). Instead of only maximizing the log-likelihood of the ground-truth output as in ML, RAML attempts to maximize the expected log-likelihood of all possible candidate outputs w.r.t. the *exponentiated payoff distribution*, which is defined as the normalized exponentiated reward. By incorporating task-specific reward into the payoff distribution, RAML combines the computational efficiency of ML with the conceptual advantages of reinforcement

learning (RL) algorithms that optimize the expected reward (Ranzato et al., 2016; Bahdanau et al., 2017). Simple as RAML appears to be, its empirical success has piqued interest in analyzing and justifying RAML from both theoretical and empirical perspectives. In their pioneering work, Norouzi et al. (2016) showed that both RAML and RL optimize the KL divergence between the exponentiated payoff distribution and model distribution, but in opposite directions. Moreover, when applied to log-linear model, RAML can also be shown to be equivalent to the softmax-margin training method (Gimpel & Smith, 2010; Gimpel, 2012). Nachum et al. (2016) applied the payoff distribution to improve the exploration properties of policy gradient for model-free reinforcement learning.

Despite these efforts, the theoretical properties of RAML, especially the interpretation and behavior of the exponentiated payoff distribution, have largely remained under-studied (§2). First, RAML attempts to match the model distribution with the heuristically designed exponentiated payoff distribution whose behavior has largely remained under-appreciated, resulting in a non-intuitive asymptotic property. Second, there is no direct theoretical proof showing that RAML can deliver a prediction function better than ML. Third, no attempt (to our best knowledge) has been made to further improve RAML from the algorithmic and practical perspectives.

In this paper, we attempt to resolve the above-mentioned under-studied problems by providing an theoretical interpretation of RAML. Our contributions are three-fold: (1) Theoretically, we introduce the framework of *softmax Q-distribution estimation*, through which we are able to interpret the role the payoff distribution plays in RAML (§3). Specifically, the softmax Q-distribution serves as a smooth approximation to the Bayes decision boundary. By comparing the payoff distribution with this softmax Q-distribution, we show that RAML approximately estimates the softmax Q-distribution, therefore approximating the Bayes decision rule. Hence, our theoretical results provide an explanation of what distribution RAML asymptotically models, and why the prediction function provided by RAML outperforms the one provided by ML. (2) Algorithmically, we further propose softmax Q-distribution maximum likelihood (SQDML) which improves RAML by achieving the exact Bayes decision boundary asymptotically. (3) Experimentally, through one experiment using synthetic data on multi-class classification and one using real data on image captioning, we verify our theoretical analysis, showing that SQDML is consistently as good or better than RAML on the task-specific metrics we desire to optimize. Additionally, through three structured prediction tasks in natural language processing (NLP) with rewards defined on sequential (named entity recognition), tree-based (dependency parsing) and complex irregular structures (machine translation), we deepen the empirical analysis of Norouzi et al. (2016), showing that RAML consistently leads to improved performance over ML on task-specific metrics, while ML yields better exact match accuracy (§4).

## 2    BACKGROUND

### 2.1    NOTATIONS

Throughout we use uppercase letters for random variables (and occasionally for matrices as well), and lowercase letters for realizations of the corresponding random variables. Let $X \in \mathcal{X}$ be the input, and $Y \in \mathcal{Y}$ be the desired structured output, e.g., in machine translation $X$ and $Y$ are French and English sentences, resp. We assume that the set of all possible outputs $\mathcal{Y}$ is finite. For instance, in machine translation all English sentences are up to a maximum length. $r(y, y^*)$ denotes the task-specific reward function (e.g., BLEU score) which evaluates a predicted output $y$ against the ground-truth $y^*$.

Let $P$ denote the true distribution of the data, i.e., $(X, Y) \sim P$, and $D = \{(x_i, y_i)\}_{i=1}^n$ be our training samples, where $\{x_i, i = 1, \ldots, n\}$ (resp. $y_i$) are usually i.i.d. samples of $X$ (resp. $Y$). Let $\mathcal{P} = \{P_\theta : \theta \in \Theta\}$ denote a parametric statistical model indexed by parameter $\theta \in \Theta$, where $\Theta$ is the parameter space. Some widely used parametric models are conditional log-linear models (Lafferty et al., 2001) and deep neural networks (Sutskever et al., 2014) (details in Appendix D.2). Once the parametric statistical model is learned, given an input $x$, model inference (a.k.a. decoding) is performed by finding an output $y^*$ achieving the highest conditional probability:

$$y^* = \underset{y \in \mathcal{Y}}{\operatorname{argmax}} \, P_{\hat{\theta}}(y|x) \tag{1}$$

where $\hat{\theta}$ is the set of parameters learned on training data $D$.

## 2.2 MAXIMUM LIKELIHOOD

Maximum likelihood minimizes the negative log-likelihood of the parameters given training data:

$$\hat{\theta}_{\text{ML}} = \underset{\theta \in \Theta}{\operatorname{argmin}} \sum_{i=1}^{n} - \log P_\theta(y_i|x_i) = \underset{\theta \in \Theta}{\operatorname{argmin}} \mathbb{E}_{\tilde{P}(X)}[\text{KL}(\tilde{P}(\cdot|X)||P_\theta(\cdot|X))] \qquad (2)$$

where $\tilde{P}(X)$ and $\tilde{P}(\cdot|X)$ is derived from the empirical distribution of training data $D$:

$$\tilde{P}(X = x, Y = y) = \frac{1}{n} \sum_{i=1}^{n} \mathbb{I}(x_i = x, y_i = y) \qquad (3)$$

and $\mathbb{I}(\cdot)$ is the indicator function. From (2), ML attempts to learn a conditional model distribution $P_{\hat{\theta}_{\text{ML}}}(\cdot|X = x)$ that is as close to the conditional empirical distribution $\tilde{P}(\cdot|X = x)$ as possible, for each $x \in \mathcal{X}$. Theoretically, under certain regularity conditions (Wasserman, 2013), asymptotically as $n \to \infty$, $P_{\hat{\theta}_{\text{ML}}}(\cdot|X = x)$ converges to the true distribution $P(\cdot|X = x)$, since $\tilde{P}(\cdot|X = x)$ converges to $P(\cdot|X = x)$ for each $x \in \mathcal{X}$.

## 2.3 REWARD AUGMENTED MAXIMUM LIKELIHOOD

As proposed in Norouzi et al. (2016), RAML incorporates task-specific rewards by re-weighting the log-likelihood of each possible candidate output proportionally to its exponentiated scaled reward:

$$\hat{\theta}_{\text{RAML}} = \underset{\theta \in \Theta}{\operatorname{argmin}} \sum_{i=1}^{n} \left\{ - \sum_{y \in \mathcal{Y}} q(y|y_i; \tau) \log P_\theta(y|x_i) \right\} \qquad (4)$$

where the reward information is encoded by the *exponentiated payoff distribution* with the temperature $\tau$ controlling it smoothness

$$q(y|y^*; \tau) = \frac{\exp(r(y, y^*)/\tau)}{\sum_{y' \in \mathcal{Y}} \exp(r(y', y^*)/\tau)} = \frac{\exp(r(y, y^*)/\tau)}{Z(y^*; \tau)} \qquad (5)$$

Norouzi et al. (2016) showed that (4) can be re-expressed in terms of KL divergence as follows:

$$\hat{\theta}_{\text{RAML}} = \underset{\theta \in \Theta}{\operatorname{argmin}} \mathbb{E}_{\tilde{P}(X,Y)}[\text{KL}(q(\cdot|Y; \tau)||P_\theta(\cdot|X))] \qquad (6)$$

where $\tilde{P}$ is the empirical distribution in (3).

**Discussion** As discussed in Norouzi et al. (2016), the globally optimal solution of RAML is achieved when the learned model distribution matches the exponentiated payoff distribution, i.e., $P_{\hat{\theta}_{\text{RAML}}}(\cdot|X = x) = q(\cdot|Y = y; \tau)$ for each $(x, y) \in D$ with some fixed value of $\tau$. It makes the payoff distribution appear to be the "target" that the model is attempting to learn. Though both $P_{\hat{\theta}_{\text{RAML}}}(\cdot|X = x)$ and $q(\cdot|Y = y; \tau)$ are distributions defined over the output space $\mathcal{Y}$, the latter is conditioned on the output $Y$ which appears to serve as ground-truth but is sampled from data distribution $P$. This makes the behavior of RAML attempting to match them unintuitive; Specifically, supposing that in the training data there exist two training instances with the same input but different outputs, i.e., $(x, y), (x, y') \in D$. Then $P_{\hat{\theta}_{\text{RAML}}}(\cdot|X = x)$ has two "targets" $q(\cdot|Y = y; \tau)$ and $q(\cdot|Y = y'; \tau)$, making it unclear what distribution $P_{\hat{\theta}_{\text{RAML}}}(\cdot|X = x)$ asymptotically converges to.

**Open Problems in RAML** Based on the discussion above, we identify two open issues in the theoretical interpretation of RAML: i) What is the (asymptotically) globally optimal solution of RAML, i.e. what distribution $P_{\hat{\theta}_{\text{RAML}}}(\cdot|X = x)$ (asymptotically) converges to; ii) There is no rigorous theoretical evidence showing that generating from $P_{\hat{\theta}_{\text{RAML}}}(y|x)$ yields a better prediction function than generating from $P_{\hat{\theta}_{\text{ML}}}(y|x)$.

To our best knowledge, no attempt has been made to *theoretically* address these problems. The main goal of this work is to theoretically analyze the properties of RAML, in hope that we may eventually better understand it by answering these questions and further improve it by proposing new training framework. To this end, in the next section we introduce a softmax Q-distribution estimation framework, facilitating our later analysis.

# 3 SOFTMAX Q-DISTRIBUTION ESTIMATION

With the end goal of theoretically interpreting RAML in mind, in this section we present the softmax Q-distribution estimation framework. We first provide background on Bayesian decision theory (§3.1) and softmax approximation of deterministic distributions (§3.2). Then, we propose the softmax Q-distribution (§3.3), and establish the framework of estimating the softmax Q-distribution from training data, called *softmax Q-distribution maximum likelihood* (SQDML, §3.4). In §3.5, we analyze SQDML, which is central in linking RAML and softmax Q-distribution estimation.

## 3.1 BAYESIAN DECISION THEORY

Bayesian decision theory is a fundamental statistical approach to the problem of pattern classification, which quantifies the trade-offs between various classification decisions using the probabilities and rewards (losses) that accompany such decisions.

Based on the notations setup in §2.1, let $\mathcal{H}$ denote all the possible prediction functions from input to output space, i.e., $\mathcal{H} = \{h : \mathcal{X} \to \mathcal{Y}\}$. Then, the *expected reward* of a prediction function $h$ is:

$$R(h) = \mathbb{E}_{P(X,Y)}[r(h(X), Y)] \tag{7}$$

where $r(\cdot, \cdot)$ is the reward function accompanied with the structured prediction task.

Bayesian decision theory states that the global maximum of $R(h)$, i.e., the optimal expected prediction reward is achieved when the prediction function is the so-called *Bayes decision rule*:

$$h^*(x) = \underset{y \in \mathcal{Y}}{\operatorname{argmax}} \mathbb{E}_{P(Y|X=x)}[r(y, Y)] = \underset{y \in \mathcal{Y}}{\operatorname{argmax}} R(y|x) \tag{8}$$

where $R(y|x) = \mathbb{E}_{P(Y|X=x)}[r(y, Y)]$ is called the *conditional reward*. Thus, the Bayes decision rule states that to maximize the overall reward, compute the conditional reward for each output $y \in \mathcal{Y}$ and then select the output $y$ for which $R(y|x)$ is maximized.

Importantly, when the reward function is the indicator function, i.e., $\mathbb{I}(y = y')$, the Bayes decision rule reduces to a specific instantiation called the *Bayes classifier*:

$$h^c(x) = \underset{y \in \mathcal{Y}}{\operatorname{argmax}} P(y|X = x) \tag{9}$$

where $P(Y|X = x)$ is the true conditional distribution of data defined in §2.1.

In §2.2, we see that ML attempts to learn the true distribution $P$. Thus, in the optimal case, decoding from the distribution learned with ML, i.e., $P_{\hat{\theta}_{\mathrm{ML}}}(Y|X = x)$, produces the Bayes classifier $h^c(x)$, but not the more general Bayes decision rule $h^*(x)$. In the rest of this section, we derive a theoretical proof showing that decoding from the distribution learned with RAML, i.e., $P_{\hat{\theta}_{\mathrm{RAML}}}(Y|X = x)$ approximately achieves $h^*(x)$, illustrating why RAML yields a prediction function with improved performance towards the optimized reward function $r(\cdot, \cdot)$ over ML.

## 3.2 SOFTMAX APPROXIMATION OF DETERMINISTIC DISTRIBUTIONS

Aimed at providing a smooth approximation of the Bayes decision boundary determined by the Bayes decision rule in (8), we first describe a widely used approximation of deterministic distributions using the softmax function.

Let $\mathcal{F} = \{f_k : k \in \mathcal{K}\}$ denote a class of functions, where $f_k : \mathcal{X} \to \mathbb{R}, \forall k \in \mathcal{K}$. We assume that $\mathcal{K}$ is finite. Then, we define the random variable $Z = \operatorname{argmax}_{k \in \mathcal{K}} f_k(X)$ where $X \in \mathcal{X}$ is our input random variable. Obviously, Z is deterministic when X is given, i.e.,

$$P(Z = z|X = x) = \begin{cases} 1, & \text{if } z = \underset{k \in \mathcal{K}}{\operatorname{argmax}} f_k(x) \\ 0, & \text{otherwise.} \end{cases} \tag{10}$$

for each $z \in \mathcal{K}$ and $x \in \mathcal{X}$.

The softmax function provides a smooth approximation of the point distribution in (10), with a temperature parameter, $\tau > 0$, serving as a hyper-parameter that controls the smoothness of the

approximating distribution around the target one:

$$Q(Z = z | X = x; \tau) = \frac{\exp(f_z(x)/\tau)}{\sum\limits_{k \in \mathcal{K}} \exp(f_k(x)/\tau)} \tag{11}$$

It should be noted that at $\tau \to 0$, the distribution $Q$ reduces to the original deterministic distribution $P$ in (10), and in the limit as $\tau \to \infty$, $Q$ is equivalent to the uniform distribution $\text{Unif}(\mathcal{K})$.

### 3.3 Softmax Q-distribution

We are now ready to propose the *softmax Q-distribution*, which is central in revealing the relationship between RAML and Bayes decision rule. We first define random variable $Z = h^*(X) = \text{argmax}_{y \in \mathcal{Y}} \mathbb{E}_{P(Y|X)}[r(y, Y)]$. Then, $Z$ is deterministic given $X$, and according to (11), we define the softmax Q-distribution to approximate the conditional distribution of $Z$ given $X$:

$$Q(Z = z | X = x; \tau) = \frac{\exp\left(\mathbb{E}_{P(Y|X=x)}[r(z, Y)]/\tau\right)}{\sum\limits_{y \in \mathcal{Y}} \exp\left(\mathbb{E}_{P(Y|X=x)}[r(y, Y)]/\tau\right)} \tag{12}$$

for each $x \in \mathcal{X}$ and $z \in \mathcal{Y}$.[1] Importantly, one can verify that decoding from the softmax Q-distribution provides us with the Bayes decision rule,

$$h(x) = \underset{y \in \mathcal{Y}}{\text{argmax}}\, Q(y | x; \tau) = \underset{y \in \mathcal{Y}}{\text{argmax}}\, \mathbb{E}_{P(Y|X=x)}[r(y, Y)] = h^*(x) \tag{13}$$

with any value of $\tau > 0$.

### 3.4 Softmax Q-distribution Maximum Likelihood

Because making predictions according to the softmax Q-distribution is equivalent to the Bayes decision rule, we would like to construct a (parametric) statistical model $\mathcal{P}$ to directly model the softmax Q-distribution in (12), similarly to how ML models the true data distribution $P$. We call this framework *softmax Q-distribution maximum likelihood (SQDML)*. This framework is model-agnostic, so any probabilistic model used in ML such as conditional log-linear models and deep neural networks, can be directly applied to modeling the softmax Q-distribution.

Suppose that we use a parametric statistical model $\mathcal{P} = \{P_\theta : \theta \in \Theta\}$ to model the softmax Q-distribution. In order to learn "optimal" parameters $\theta$ from training data $D = \{(x_i, y_i)\}_{i=1}^n$, an intuitive and well-motivated objective function is the KL-divergence between the empirical conditional distribution of $Q(\cdot | X)$, denoted as $\tilde{Q}(\cdot | X)$, and the model distribution $P_\theta(\cdot | X)$:

$$\hat{\theta}_{\text{SQDML}} = \underset{\theta \in \Theta}{\text{argmin}}\, \mathbb{E}_{\tilde{Q}(X)}[\text{KL}(\tilde{Q}(\cdot | X) || P_\theta(\cdot | X))] \tag{14}$$

We can directly set $\tilde{Q}(X) = \tilde{P}(X)$, which leaves the problem of defining the empirical conditional distribution $\tilde{Q}(Z | X)$. Before defining $\tilde{Q}(Z | X)$, we first note that if the defined empirical distribution $\tilde{Q}(X, Z)$ asymptotically converges to the true Q-distribution $Q(X, Z)$, the learned model distribution $P_{\hat{\theta}_{\text{SQDML}}}(\cdot | X = x)$ converges to $Q(\cdot | X = x)$. Therefore, decoding from $P_{\hat{\theta}_{\text{SQDML}}}(\cdot | X = x)$ ideally achieves the Bayes decision rule $h^*(x)$.

A straightforward way to define $\tilde{Q}(Z | X = x)$ is to use the empirical distribution $\tilde{P}(Y | X = x)$:

$$\tilde{Q}(Z = z | X = x) = \frac{\exp\left(\mathbb{E}_{\tilde{P}(Y|X=x)}[r(z, Y)]/\tau\right)}{\sum\limits_{y \in \mathcal{Y}} \exp\left(\mathbb{E}_{\tilde{P}(Y|X=x)}[r(y, Y)]/\tau\right)} \tag{15}$$

where $\tilde{P}$ is the empirical distribution of $P$ defined in (3). Asymptotically as $n \to \infty$, $\tilde{P}$ converges to $P$. Thus, $\tilde{Q}$ asymptotically converges to $Q$.

---

[1]In the following derivations we omit $\tau$ in $Q(Z | X; \tau)$ for simplicity when there is no ambiguity.

Unfortunately, the empirical distribution $\tilde{Q}$ (15) is not efficient to compute, since the expectation term is inside the exponential function (See appendix D.2 for approximately learning $\hat{\theta}_{\text{SQDML}}$ in practice). This leads us to seek an approximation of the softmax Q-distribution and its corresponding empirical distribution. Here we propose the following $Q'$ distribution to approximate the softmax Q-distribution $Q$ defined in (12):

$$Q'(Z = z|X = x; \tau) = \mathbb{E}_{P(Y|X=x)}\left[ \frac{\exp\left(r(z, Y)/\tau\right)}{\sum\limits_{y \in \mathcal{Y}} \exp\left(r(y, Y)/\tau\right)} \right] \tag{16}$$

where we move the expectation term outside the exponential function. Then, the corresponding empirical distribution of $Q'(X, Z)$ can be written in the following form:

$$\tilde{Q}'(X = x, Z = z) = \frac{1}{n} \sum_{i=1}^{n} \left\{ \sum_{y \in \mathcal{Y}} \frac{\exp(r(z, y)/\tau)}{\sum\limits_{y' \in \mathcal{Y}} \exp(r(y', y)/\tau)} \mathbb{I}(x_i = x, y_i = y) \right\} \tag{17}$$

Approximating $\tilde{Q}(X, Z)$ with $\tilde{Q}'(X, Z)$, and plugging (17) into the RHS in (14), we have:

$$\begin{aligned}
\hat{\theta}_{\text{SQDML}} &\approx \underset{\theta \in \Theta}{\operatorname{argmin}} \, \mathbb{E}_{\tilde{Q}'(X)}[\text{KL}(\tilde{Q}'(\cdot|X) \| P_\theta(\cdot|X))] \\
&= \underset{\theta \in \Theta}{\operatorname{argmin}} \sum_{i=1}^{n} \left\{ - \sum_{y \in \mathcal{Y}} q(y|y_i; \tau) \log P_\theta(y|x_i) \right\} = \hat{\theta}_{\text{RAML}}
\end{aligned} \tag{18}$$

where $q(y|y^*; \tau)$ is the exponentiated payoff distribution of RAML in (5).

Equation (18) states that RAML is an approximation of our proposed SQDML by approximating $\tilde{Q}$ with $\tilde{Q}'$. Interestingly and mostly in practice, when the input is unique in the training data, i.e., $\nexists (x_1, y_1), (x_2, y_2) \in D$, s.t. $x_1 = x_2 \wedge y_1 \neq y_2$, we have $\tilde{Q} = \tilde{Q}'$, resulting in $\hat{\theta}_{\text{SQDML}} = \hat{\theta}_{\text{RAML}}$. It states that the estimated distribution $P_{\hat{\theta}_{\text{SQDML}}}$ and $P_{\hat{\theta}_{\text{RAML}}}$ are exactly the same when the input $x$ is unique in the training data, since the empirical distributions $\tilde{Q}$ and $\tilde{Q}'$ estimated from the training data are the same.

## 3.5 Analysis and Discussion of SQDML

In §3.4, we provided a theoretical interpretation of RAML by establishing the relationship between RAML and SQDML. In this section, we try to answer the questions of RAML raised in §2.3 using this interpretation and further analyze the level of approximation from the softmax Q-distribution $Q$ in (13) to $Q'$ in (16) by proving a upper bound of the approximation error.

Let's first use our interpretation to answer the open questions regarding RAML in §2.3. First, instead of optimizing the KL divergence between the artificially designed exponentiated payoff distribution and the model distribution, RAML in our formulation approximately matches model distribution $P_\theta(\cdot|X = x)$ with the softmax Q-distribution $Q(\cdot|X = x; \tau)$. Second, based on our interpretation, asymptotically as $n \to \infty$, RAML learns a distribution that converges to $Q'(\cdot)$ in (16), and therefore *approximately* converges to the softmax Q-distribution. Third, as mentioned in §3.3, generating from the softmax Q-distribution produces the Bayes decision rule, which theoretically outperforms the prediction function from ML, w.r.t. the expected reward.

It is necessary to mention that both RAML and SQDML are trying to learn distributions, decoding from which (approximately) delivers the Bayes decision rule. There are other directions that can also achieve the Bayes decision rule, such as minimum Bayes risk decoding (Kumar & Byrne, 2004), which attempts to estimate the Bayes decision rule directly by computing expectation w.r.t the data distribution learned from training data.

So far our discussion has concentrated on the theoretical interpretation and analysis of RAML, without any concerns for how well $Q'(X, Z)$ approximates $Q(X, Z)$. Now, we characterize the approximating error by proving a upper bound of the KL divergence between them:

**Theorem 1.** *Given the input and output random variable $X \in \mathcal{X}$ and $Y \in \mathcal{Y}$ and the data distribution $P(X, Y)$. Suppose that the reward function is bounded $0 \leq r(y, y^*) \leq R$. Let $Q(Z|X; \tau)$ and $Q'(Z|X; \tau)$ be the softmax Q-distribution and its approximation defined in (12) and (16). Assume that $Q(X) = Q'(X) = P(X)$. Then,*

$$\text{KL}(Q(\cdot, \cdot) \| Q'(\cdot, \cdot)) \leq 2R/\tau \tag{19}$$

From Theorem 1 (proof in Appendix A.1) we observe that the level of approximation mainly depends on two factors: the upper bound of the reward function ($R$) and the temperature parameter $\tau$. In practice, $R$ is often less than or equal to 1, when metrics like accuracy or BLEU are applied.

It should be noted that, at one extreme when $\tau$ becomes larger, the approximation error tends to be zero. At the same time, however, the softmax Q-distribution becomes closer to the uniform distribution $\mathrm{Unif}(\mathcal{Y})$, providing less information for prediction. Thus, in practice, it is necessary to consider the trade-off between approximation error and predictive power.

What about the other extreme — $\tau$ "as close to zero as possible"? With suitable assumptions about the data distribution $P$, we can characterize the approximating error by using the same KL divergence:

**Theorem 2.** *Suppose that the reward function is bounded $0 \leq r(y, y^*) \leq R$, and $\forall y' \neq y$, $r(y, y) - r(y', y) \geq \gamma R$ where $\gamma \in (0, 1)$ is a constant. Suppose additionally that, like a sub-Gaussian, for every $x \in \mathcal{X}$, $P(Y|X = x)$ satisfies the exponential tail bound w.r.t. $r$ — that is, for each $x \in \mathcal{X}$, there exists a unique $y^* \in \mathcal{Y}$ such that for every $t \in [0, 1)$*

$$P(r(y^*, y^*) - r(Y, y^*) \geq tR|X = x) \leq e^{-c\frac{t^2}{(1-t)^2}} \tag{20}$$

*where $c$ is a distribution-dependent constant. Assume that $Q(X) = Q'(X) = P(X)$. Denote $b = \frac{\gamma^2}{(1-\gamma)^2}$. Then, as $\tau \to 0$,*

$$\mathrm{KL}(Q(\cdot, \cdot)\|Q'(\cdot, \cdot)) \leq \frac{1}{1 + e^{cb}}. \tag{21}$$

Theorem 2 (proof in Appendix A.2) indicates that RAML can also achieve little approximating error when $\tau$ is close to zero.

## 4 EXPERIMENTS

In this section, we performed two sets of experiments to verity our theoretical analysis of the relation between SQDML and RAML. As discussed in §3.4, RAML and SQDML deliver the same predictions when the input $x$ is unique in the data. Thus, in order to compare SQDML against RAML, the first set of experiments are designed on two data sets in which $x$ is not unique — synthetic data for cost-sensitive multi-class classification, and the MSCOCO benchmark dataset (Chen et al., 2015) for image captioning. To further confirm the advantages of RAML (and SQDML) over ML, and thus the necessity for better theoretical understanding, we performed the second set of experiments on three structured prediction tasks in NLP. In these cases SQDML reduces to RAML, as the input is unique in these three data sets.

### 4.1 EXPERIMENTS ON SQDML

#### 4.1.1 COST-SENSITIVE MULTI-CLASS CLASSIFICATION

First, we perform experiments on synthetic data for cost-sensitive multi-class classification designed to demonstrate that RAML learns a distribution approximately producing the Bayes decision rule, which is asymptotically the prediction function delivered by SQDML.

The synthetic data set is for a 4-class classification task, where $x \in \mathcal{X} = [-1, +1] \times [-1, +1] \subset \mathcal{R}^2$, and $y \in \mathcal{Y} = \{0, 1, 2, 3\}$. We define four base points, one for each class:

$$\begin{bmatrix} x_0 \\ x_1 \\ x_2 \\ x_3 \end{bmatrix} = \begin{bmatrix} +1 & +1 \\ +1 & -1 \\ -1 & -1 \\ -1 & +1 \end{bmatrix}$$

For data generation, the distribution $P(X)$ is the uniform distribution on $\mathcal{X}$, and the log form of the conditional distribution $P(Y|X = x)$ for each $x \in \mathcal{X}$ is proportional to the negative distance of each base point:

$$\log P(Y = y|X = x) \propto -d(x, x_y), \text{ for } y \in \{0, 1, 2, 3\} \tag{22}$$

where $d(\cdot, \cdot)$ is the Euclidean distance between two points. To generate training data, we first draw 1 million inputs $x$ from $P(X)$. Then, we independently generate 10 outputs y from $P(Y|X = x)$ for

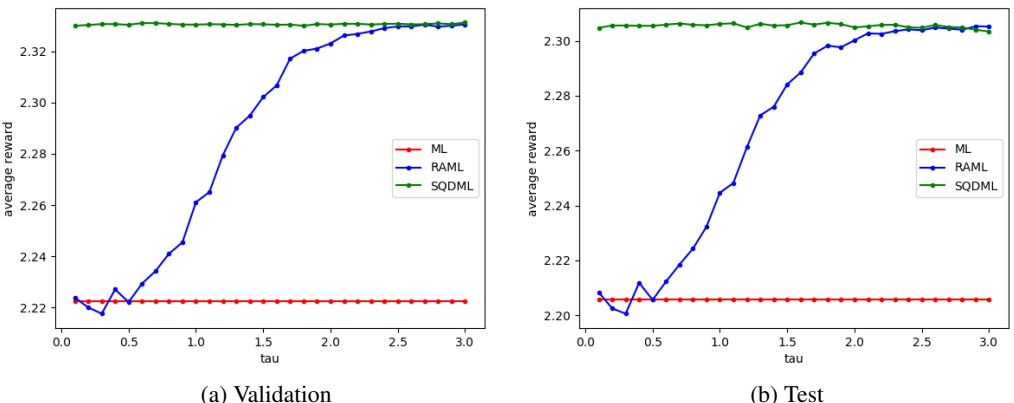

Figure 1: Average reward relative to the temperature parameter $\tau$, ranging from 0.1 to 3.0, on validation and test sets, respectively.

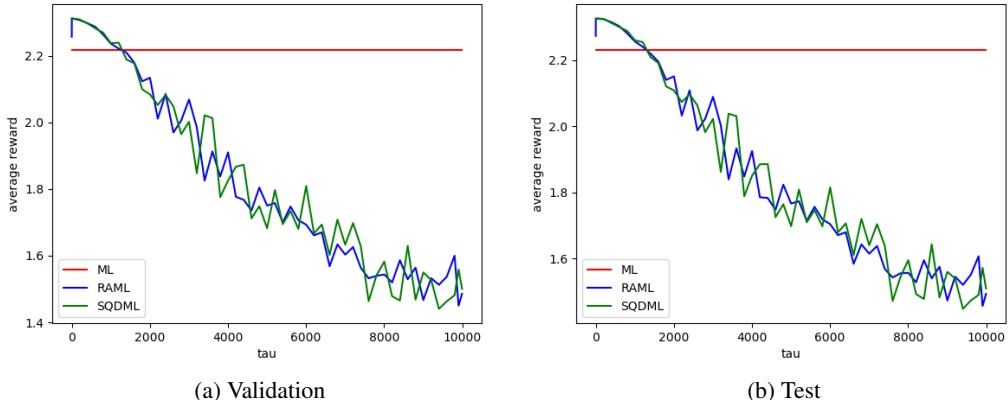

Figure 2: Average reward relative to a wide range of $\tau$ (from 1.0 to 10,000) on validation and test sets, respectively.

each $x$ to build a data set with multiple references. Thus, the total number of training instances is 10 million. For validation and test data, we independently generate 0.1 million pairs of $(x, y)$ from $P(X, Y)$, respectively.

The model we used is a feed-forward (dense) neural networks with 2 hidden layers, each of which has 8 units. Optimization is performed with mini-batch stochastic gradient descent (SGD) with learning rate 0.1 and momentum 0.9. Each model is trained with 100 epochs and we apply early stopping (Caruana et al., 2001) based on performance on validation sets.

The reward function $r(\cdot, \cdot)$ is designed to distinguish the four classes. For "correct" predictions, the specific reward values assigned for the four classes are:

$$\begin{bmatrix} r(0,0) \\ r(1,1) \\ r(2,2) \\ r(3,3) \end{bmatrix} = \begin{bmatrix} e^{2.0} \\ e^{1.6} \\ e^{1.2} \\ e^{1.1} \end{bmatrix}$$

For "wrong" predictions, rewards are always zero, i.e. $r(y, y^*) = 0$ when $y \neq y^*$.

Figure 1 depicts the effect of varying the temperature parameter $\tau$ on model performance, ranging from 0.1 to 3.0 with step 0.1. For each fixed $\tau$, we report the mean performance over 5 repetitions. Figure 1 shows the averaged rewards obtained as a function of $\tau$ on both validation and test datasets

| $\tau$ | RAML | | SQDML | | $\tau$ | RAML | | SQDML | |
|---|---|---|---|---|---|---|---|---|---|
| | Reward | BLEU | Reward | BLEU | | Reward | BLEU | Reward | BLEU |
| $\tau = 0.80$ | 10.77 | 27.02 | 10.82 | 27.08 | $\tau = 1.00$ | 10.84 | 27.26 | 10.82 | 27.03 |
| $\tau = 0.85$ | 10.81 | 27.27 | 10.78 | 26.92 | $\tau = 1.05$ | 10.82 | 27.29 | 10.80 | 27.20 |
| $\tau = 0.90$ | **10.88** | **27.62** | **10.91** | **27.54** | $\tau = 1.10$ | 10.74 | 26.89 | 10.78 | 26.98 |
| $\tau = 0.95$ | 10.82 | 27.33 | 10.79 | 27.02 | $\tau = 1.15$ | 10.77 | 27.01 | 10.72 | 26.66 |

Table 1: Average **Reward** (sentence-level BLEU) and corpus-level **BLEU** (standard evaluation metric) scores for image captioning task with different $\tau$.

of ML, RAML and SQDML, respectively. From Figure 1 we can see that when $\tau$ increases, the performance gap between SQDML and RAML keeps decreasing, indicting that RAML achieves better approximation to SQDML. This evidence verities the statement in Theorem 1 that the approximating error between RAML and SQDML decreases when $\tau$ continues to grow.

The results in Figure 1 raise a question: does larger $\tau$ necessarily yield better performance for RAML? To further illustrate the effect of $\tau$ on model performance of RAML and SQDML, we perform experiments with a wide range of $\tau$ — from 1 to 10,000 with step 200. We also repeat each experiment 5 times. The results are shown in Figure 2. We see that the model performance (average reward), however, has not kept growing with increasing $\tau$. As discussed in §3.5, the softmax Q-distribution becomes closer to the uniform distribution when $\tau$ becomes larger, making it less expressive for prediction. Thus, when applying RAML in practice, considerations regarding the trade-off between approximating error and predictive power of model are needed. More details, results and analysis of the conducted experiments are provided in Appendix B.

### 4.1.2 IMAGE CAPTIONING WITH MULTIPLE REFERENCES

Second, to show that optimizing toward our proposed SQDML objective yields better predictions than RAML on real-world structured prediction tasks, we evaluate on the MSCOCO image captioning dataset. This dataset contains 123,000 images, each of which is paired with as least five manually annotated captions. We follow the offline evaluation setting in (Karpathy & Li, 2015), and reserve 5,000 images for validation and testing, respectively. We implemented a simple neural image captioning model using a pre-trained VGGNet as the encoder and a Long Short-Term Memory (LSTM) network as the decoder. Details of the experimental setup are in Appendix C.

As in §4.1.1, for the sake of comparing SQDML with RAML to verify our theoretical analysis, we use the average reward as the performance measure by simply defining the reward as pairwise sentence level BLEU score between model's prediction and each reference caption[2], though the standard benchmark metric commonly used in image captioning (e.g., corpus-level BLEU-4 score) is not simply defined as averaging over the pairwise rewards between prediction and reference captions.

We use stochastic gradient descent to optimize the objectives for SQDML (14) and RAML (4). However, the denominators of the softmax-Q distribution for SQDML $\tilde{Q}(Z|X;\tau)$ (15) and the payoff distribution for RAML $q(y|y^*;\tau)$ (5) contain summations over intractable exponential hypotheses space $\mathcal{Y}$. We therefore propose a simple heuristic approach to approximate the denominator by restricting the exponential space $\mathcal{Y}$ using a fixed set $\mathcal{S}$ of sampled targets, i.e., $\mathcal{Y} \approx \mathcal{S}$. Approximating the intractable hypotheses space using sampling is not new in structured prediction, and has been shown effective in optimizing neural structured prediction models (Shen et al., 2016). Specifically, the sampled candidate set $\mathcal{S}$ is constructed by (i) including each ground-truth reference $y^*$ into $\mathcal{S}$; and (ii) uniformly replacing an $n$-gram ($n \in \{1, 2, 3\}$) in one (randomly sampled) reference $y^*$ with a randomly sampled $n$-gram. We refer to this approach as $n$-*gram replacement*. We provide more details of the training procedure in Appendix C.

Table 1 lists the results. We evaluate on both the average reward and the benchmark metric (corpus-level BLEU-4). We also tested on a vanilla ML baseline, which achieves 10.71 average reward and 26.91 corpus-level BLEU. Both SQDML and RAML outperform ML according to the two metrics. Interestingly, comparing SQDML with RAML we did not observe a significant improvement of

---

[2]Not that this is different from standard multi-reference sentence-level BLEU, which counts n-gram matches w.r.t. all sentences then uses these sufficient statistics to calculate a final score.

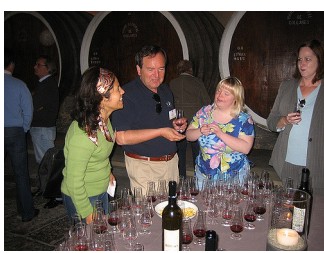

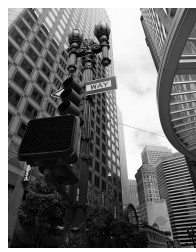

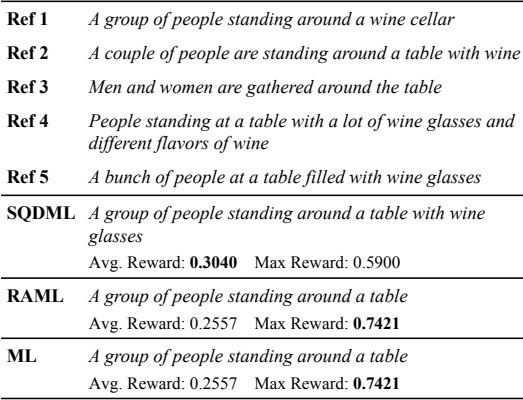

| | |
|---|---|
| **Ref 1** | *A group of people standing around a wine cellar* |
| **Ref 2** | *A couple of people are standing around a table with wine* |
| **Ref 3** | *Men and women are gathered around the table* |
| **Ref 4** | *People standing at a table with a lot of wine glasses and different flavors of wine* |
| **Ref 5** | *A bunch of people at a table filled with wine glasses* |
| **SQDML** | *A group of people standing around a table with wine glasses* 
 Avg. Reward: **0.3040**   Max Reward: 0.5900 |
| **RAML** | *A group of people standing around a table* 
 Avg. Reward: 0.2557   Max Reward: **0.7421** |
| **ML** | *A group of people standing around a table* 
 Avg. Reward: 0.2557   Max Reward: **0.7421** |

| | |
|---|---|
| **Ref 1** | *A one way sign that is on a pole* |
| **Ref 2** | *A black and white picture of a traffic signal in a city* |
| **Ref 3** | *A black and white image of some buildings and a street light* |
| **Ref 4** | *Intersection with traffic signals in large metropolitan area* |
| **Ref 5** | *Traffic lights in front of large buildings with a one way sign* |
| **SQDML** | *A black and white photo of a street sign on a pole* 
 Avg. Reward: **0.1424**   Max Reward: 0.2620 |
| **RAML** | *A black and white photo of a traffic light* 
 Avg. Reward: 0.1409   Max Reward: **0.3093** |
| **ML** | *A black and white photo of a street sign* 
 Avg. Reward: 0.1253   Max Reward: 0.2643 |

Figure 3: Testing examples from MSCOCO image captioning task

average reward. We hypothesize that this is due to the fact that the reference captions for each image are largely different, making it highly non-trivial for the model to predict a "consensus" caption that agrees with multiple references. As an example, we randomly sampled 300 images from the validation set and compute the averaged sentence-level BLEU between two references, which is only 10.09. Nevertheless, through case studies we still found some interesting examples, which demonstrate that SQDML is capable of generating predictions that match with multiple candidates. Figure 3 gives two examples. In the two examples, SQDML's predictions match with multiple references, registering the highest average reward. On the other hand, RAML gives sub-optimal predictions in terms of average reward since it is an approximation of SQDML. And finally for ML, since its objective is solely maximizing the reward w.r.t a single reference, it gives the lowest average reward, while achieving higher maximum reward.

## 4.2 EXPERIMENTS ON STRUCTURED PREDICTION

Norouzi et al. (2016) already evaluated the effectiveness of RAML on sequence prediction tasks of speech recognition and machine translation using neural sequence-to-sequence models. In this section, we further confirm the empirical success of RAML (and SQDML) over ML: (i) We apply RAML on three structured prediction tasks in NLP, including named entity recognition (NER), dependency parsing and machine translation (MT), using both classical feature-based log-linear models (NER and parsing) and state-of-the-art attentional recurrent neural networks (MT). (ii) Different from Norouzi et al. (2016) where edit distance is uniformly used as a surrogate training reward and the learning objective in (4) is approximated through sampling, we use task-specific rewards, defined on sequential (NER), tree-based (parsing) and complex irregular structures (MT). Specifically, instead of sampling, we apply efficient dynamic programming algorithms (NER and parsing) to directly compute the analytical solution of (4). (iii) We present further analysis comparing RAML with ML, showing that due to different learning objectives, RAML registers better results under task-specific metrics, while ML yields better exact-match accuracy.

### 4.2.1 SETUP

In this section we describe experimental setups for three evaluation tasks. We refer readers to Appendix D for dataset statistics, modeling details and training procedure.

| Method | DEV. Results | | TEST Results | | Method | DEV. Results | TEST Results |
| | Acc | F1 | Acc | F1 | | UAS | UAS |
|---|---|---|---|---|---|---|---|
| ML Baseline | 98.2 | 90.4 | 97.0 | 84.9 | ML Baseline | 91.3 | 90.7 |
| $\tau = 0.1$ | 98.3 | 90.5 | 97.0 | 85.0 | $\tau = 0.1$ | 91.0 | 90.6 |
| $\tau = 0.2$ | **98.4** | **91.2** | **97.3** | **86.0** | $\tau = 0.2$ | 91.5 | 91.0 |
| $\tau = 0.3$ | 98.3 | 90.2 | 97.1 | 84.7 | $\tau = 0.3$ | **91.7** | **91.1** |
| $\tau = 0.4$ | 98.3 | 89.6 | 97.1 | 84.0 | $\tau = 0.4$ | 91.4 | 90.8 |
| $\tau = 0.5$ | 98.3 | 89.4 | 97.1 | 83.3 | $\tau = 0.5$ | 91.2 | 90.7 |
| $\tau = 0.6$ | 98.3 | 88.9 | 97.0 | 82.8 | $\tau = 0.6$ | 91.0 | 90.6 |
| $\tau = 0.7$ | 98.3 | 88.6 | 97.0 | 82.2 | $\tau = 0.7$ | 90.8 | 90.4 |
| $\tau = 0.8$ | 98.2 | 88.5 | 96.9 | 81.9 | $\tau = 0.8$ | 90.8 | 90.3 |
| $\tau = 0.9$ | 98.2 | 88.5 | 97.0 | 82.1 | $\tau = 0.9$ | 90.7 | 90.1 |

Table 2: Token accuracy and official F1 for NER.    Table 3: UAS scores for dependency parsing.

**Named Entity Recognition (NER)**   For NER, we experimented on the English data from CoNLL 2003 shared task (Tjong Kim et al., 2003). There are four predefined types of named entities: *PERSON, LOCATION, ORGANIZATION,* and *MISC*. The dataset includes 15K training sentences, 3.4K for validation, and 3.7K for testing.

We built a linear CRF model (Lafferty et al., 2001) with the same features used in Finkel et al. (2005). Instead of using the official F1 score over complete span predictions, we use token-level accuracy as the training reward, as this metric can be factorized to each word, and hence there exists efficient dynamic programming algorithm to compute the expected log-likelihood objective in (4).

**Dependency Parsing**   For dependency parsing, we evaluate on the English Penn Treebanks (PTB) (Marcus et al., 1993). We follow the standard splits of PTB, using sections 2-21 for training, section 22 for validation and 23 for testing. We adopt the Stanford Basic Dependencies (De Marneffe et al., 2006) using the Stanford parser v3.3.0[3]. We applied the same data preprocessing procedure as in Dyer et al. (2015).

We adopt an edge-factorized tree-structure log-linear model with the same features used in Ma & Zhao (2012). We use the unlabeled attachment score (UAS) as the training reward, which is also the official evaluation metric of parsing performance. Similar as NER, the expectation in (4) can be computed deficiently using dynamic programming since UAS can be factorized to each edge.

**Machine Translation (MT)**   We tested on the German-English machine translation task in the IWSLT 2014 evaluation campaign (Cettolo et al., 2014), a widely-used benchmark for evaluating optimization techniques for neural sequence-to-sequence models. The dataset contains 153K training sentence pairs. We follow previous works (Wiseman & Rush, 2016; Bahdanau et al., 2017; Li et al., 2017) and use an attentional neural encoder-decoder model with LSTM networks. The size of the LSTM hidden states is 256. Similar as in §4.1.2, we use the sentence level BLEU score as the training reward and approximate the learning objective using $n$-gram replacement ($n \in \{1, 2, 3, 4\}$). We evaluate using standard corpus-level BLEU.

### 4.2.2   MAIN RESULTS

The results of NER and dependency parsing are shown in Table 2 and Table 3, respectively. We observed that the RAML model obtained the best results at $\tau = 0.2$ for NER, and $\tau = 0.3$ for dependency parsing. Beyond $\tau = 0.4$, RAML models get worse than the ML baseline for both the two tasks, showing that in practice selection of temperature $\tau$ is needed. In addition, the rewards we directly optimized in training (token-level accuracy for NER and UAS for dependency parsing) are more stable w.r.t. $\tau$ than the evaluation metrics (F1 in NER), illustrating that in practice, choosing a training reward that correlates well with the evaluation metric is important.

Table 4 summarizes the results for MT. We also compare our model with previous works on incorporating task-specific rewards (i.e., BLEU score) in optimizing neural sequence-to-sequence models (c.f. Table 5). Our approach, albeit simple, surprisingly outperforms previous works. Specifically,

---

[3]http://nlp.stanford.edu/software/lex-parser.shtml

| $\tau$ | S-B | C-B | $\tau$ | S-B | C-B |
|---|---|---|---|---|---|
| $\tau = 0.1$ | 28.67 | 27.42 | $\tau = 0.6$ | 29.37 | 28.49 |
| $\tau = 0.2$ | 29.44 | 28.38 | $\tau = 0.7$ | 29.52 | 28.59 |
| $\tau = 0.3$ | 29.59 | 28.40 | $\tau = 0.8$ | 29.54 | 28.63 |
| $\tau = 0.4$ | **29.80** | **28.77** | $\tau = 0.9$ | 29.48 | 28.58 |
| $\tau = 0.5$ | 29.55 | 28.45 | $\tau = 1.0$ | 29.34 | 28.40 |

Table 4: Sentence-level BLEU (**S-B**, training reward) and corpus-level BLEU (**C-B**, standard evaluation metric) scores for RAML with different $\tau$.

| Methods | ML Baseline | Proposed Model |
|---|---|---|
| Ranzato et al. (2016) | 20.10 | 21.81 |
| Wiseman & Rush (2016) | 24.03 | 26.36 |
| Li et al. (2017) | **27.90** | 28.30 |
| Bahdanau et al. (2017) | 27.56 | 28.53 |
| This Work | 27.66 | **28.77** |

Table 5: Comparison of our proposed approach with previous works. All previous methods require pre-training using an ML baseline, while RAML learns from scratch.

| | NER | | | Parsing | | MT | | |
|---|---|---|---|---|---|---|---|---|
| Metric | Acc. | F1 | E.M. | UAS | E.M. | S-B | C-B | E.M. |
| ML | 97.0 | 84.9 | 78.8 | 90.7 | **39.9** | 29.15 | 27.66 | **3.79** |
| RAML | **97.3** | **86.0** | **80.1** | **91.1** | 39.4 | **29.80** | **28.77** | 3.35 |

Table 6: Performance of ML and RAML under different metrics for the three tasks on test sets. **E.M.** refers to exact match accuracy.

all previous methods require a pre-trained ML baseline to initialize the model, while RAML learns from scratch. This suggests that RAML is easier and more stable to optimize compared with existing approaches like RL (e.g., Ranzato et al. (2016) and Bahdanau et al. (2017)), which requires sampling from the moving model distribution and suffers from high variance. Finally, we remark that RAML performs consistently better than the ML (27.66) across most temperature terms.

### 4.2.3 FURTHER COMPARISON WITH MAXIMUM LIKELIHOOD

Table 6 illustrates the performance of ML and RAML under different metrics of the three tasks. We observe that RAML outperforms ML on both the directly optimized rewards (token-level accuracy for NER, UAS for dependency parsing and sentence-level BLEU for MT) and task-specific evaluation metrics (F1 for NER and corpus-level BLEU for MT). Interestingly, we find a trend that ML gets better results on two out of the three tasks under exact match accuracy, which is the reward that ML attempts to optimize (as discussed in (9)). This is in line with our theoretical analysis, in that RAML and ML achieve better prediction functions w.r.t. their corresponding rewards they try to optimize.

## 5 CONCLUSION

In this work, we propose the framework of estimating the softmax Q-distribution from training data. Based on our theoretical analysis, asymptotically, the prediction function learned by RAML approximately achieves the Bayes decision rule. Experiments on three structured prediction tasks demonstrate that RAML consistently outperforms ML baselines.

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

APPENDIX: SOFTMAX Q-DISTRIBUTION ESTIMATION FOR STRUCTURED PREDICTION: A THEORETICAL INTERPRETATION FOR RAML

## A  SOFTMAX Q-DISTRIBUTION MAXIMUM LIKELIHOOD

### A.1  PROOF OF THEOREM 1

*Proof.* Since the reward function is bounded $0 \leq r(y, y^*) \leq R, \forall y, y* \in \mathcal{Y}$, we have:

$$1 \leq \exp(r(y, y^*)/\tau) \leq e^{R/\tau}$$

Then,

$$\frac{1}{|\mathcal{Y}|e^{R/\tau}} < \frac{1}{1 + (|\mathcal{Y}| - 1)e^{R/\tau}} \leq \frac{\exp(r(y, y^*)/\tau)}{\sum\limits_{y' \in \mathcal{Y}} \exp(r(y', y^*)/\tau)} \leq \frac{e^{R/\tau}}{|\mathcal{Y}| - 1 + e^{R/\tau}} < \frac{e^{R/\tau}}{|\mathcal{Y}|} \quad (1)$$

Now we can bound the conditional distribution $Q(z|x)$ and $Q'(z|x)$:

$$\frac{1}{|\mathcal{Y}|e^{R/\tau}} < Q(Z = z|X = x; \tau) = \frac{\exp\left(\mathbb{E}_P[r(z, Y)|X = x]/\tau\right)}{\sum\limits_{y \in \mathcal{Y}} \exp\left(\mathbb{E}_P[r(y, Y)|X = x]/\tau\right)} < \frac{e^{R/\tau}}{|\mathcal{Y}|} \quad (2)$$

and,

$$\frac{1}{|\mathcal{Y}|e^{R/\tau}} < Q'(Z = z|X = x; \tau) = \mathbb{E}_P\left[\frac{\exp\left(r(z, Y)/\tau\right)}{\sum\limits_{y \in \mathcal{Y}} \exp\left(r(y, Y)/\tau\right)}\Big| X = x\right] < \frac{e^{R/\tau}}{|\mathcal{Y}|} \quad (3)$$

Thus, $\forall x \in \mathcal{X}, z \in \mathcal{Y}$,

$$\log \frac{Q(z|x)}{Q'(z|x)} < 2R/\tau$$

To sum up, we have:

$$
\begin{aligned}
\mathrm{KL}(Q(\cdot, \cdot)\|Q'(\cdot, \cdot)) &= \sum_{x \in \mathcal{X}} Q(x) \sum_{z \in \mathcal{Y}} Q(z|x) \log \frac{Q(z|x)Q(x)}{Q'(z|x)Q'(x)} \\
&= \sum_{x \in \mathcal{X}} Q(x) \sum_{z \in \mathcal{Y}} Q(z|x) \log \frac{Q(z|x)}{Q'(z|x)} \\
&< \sum_{x \in \mathcal{X}} Q(x) \sum_{z \in \mathcal{Y}} Q(z|x) 2R/\tau \\
&= 2R/\tau
\end{aligned}
$$

$\square$

### A.2  PROOF OF THEOREM 2

**Lemma 3.** *For every* $x \in \mathcal{X}$,

$$P(Y \neq y^*|X = x) \leq e^{-cb}$$

*where* $b \triangleq \frac{\gamma^2}{(1-\gamma)^2}$.

*Proof.* From the assumption in Theorem 2 of Eq. (20) in §3.5, we have

$$P(Y \neq y^*|X = x) = P(r(y^*, y^*) - r(Y, y^*) \geq \gamma R) \leq e^{-c\frac{\gamma^2}{(1-\gamma)^2}} = e^{-cb}$$

$\square$

**Lemma 4.**

$$
\begin{aligned}
\frac{1}{1 + (|\mathcal{Y}| - 1)e^{R/\tau}} &\leq q(y|y^*) \leq \frac{1}{e^{\gamma R/\tau} + (|\mathcal{Y}| - 1)e^{-R/\tau}} &, \textit{if } y \neq y^* \\
\frac{1}{1 + (|\mathcal{Y}| - 1)e^{-\gamma R/\tau}} &\leq q(y|y^*) \leq \frac{1}{1 + (|\mathcal{Y}| - 1)e^{-R/\tau}} &, \textit{if } y = y^*
\end{aligned}
$$

*Proof.* From Eq. (1), we have

$$\frac{1}{1+(|\mathcal{Y}|-1)e^{R/\tau}} \le q(y|y^*) \le \frac{1}{1+(|\mathcal{Y}|-1)e^{-R/\tau}}$$

If $y \ne y^*$,

$$q(y|y^*) = \frac{\exp(r(y,y^*)/\tau)}{\sum\limits_{y'\in\mathcal{Y}} \exp(r(y',y^*)/\tau)} \le \frac{e^{r(y,y^*)/\tau}}{e^{r(y^*,y^*)/\tau}+(|\mathcal{Y}|-1)} \le \frac{1}{e^{\gamma R/\tau}+(|\mathcal{Y}|-1)e^{-R/\tau}}$$

If $y = y^*$,

$$q(y|y^*) = q(y^*|y^*) = \frac{1}{1+\sum\limits_{y'\ne y^*} e^{(r(y',y^*)-r(y^*,y^*))/\tau}} \le \frac{1}{1+(|\mathcal{Y}|-1)e^{-\gamma R/\tau}}$$

$\square$

**Lemma 5.**

$$\begin{array}{ccccl}
\frac{1}{1+(|\mathcal{Y}|-1)e^{R/\tau}} & \le & Q'(z|x) & \le & \frac{1}{e^{\gamma R/\tau}+(|\mathcal{Y}|-1)e^{-R/\tau}} + \frac{e^{-cb}}{1+(|\mathcal{Y}|-1)e^{-R/\tau}} \quad , \text{if } z \ne y^* \\
\frac{1-e^{-cb}}{1+(|\mathcal{Y}|-1)e^{-\gamma R/\tau}} & \le & Q'(z|x) & \le & \frac{1}{1+(|\mathcal{Y}|-1)e^{-R/\tau}} \quad , \text{if } y = y^*
\end{array}$$

*Proof.*

$$Q'(z|x) = \mathrm{E}_p[q(z|Y)] = \sum_{y\in\mathcal{Y}} p(y|x)q(z|y)$$

From Eq. (3) we have,

$$\frac{1}{1+(|\mathcal{Y}|-1)e^{R/\tau}} \le Q'(z|x) \le \frac{1}{1+(|\mathcal{Y}|-1)e^{-R/\tau}}$$

$\square$

If $z \ne y^*$,

$$Q'(z|x) = p(y^*|x)q(z|y^*) + \sum_{y\ne y^*} p(y|x)q(z|y) \le q(z|y^*) + \frac{1}{1+(|\mathcal{Y}|-1)e^{-R/\tau}}P(Y\ne y^*|X=x)$$

From Lemma 3 and Lemma 4,

$$Q'(z|x) \le \frac{1}{e^{\gamma R/\tau}+(|\mathcal{Y}|-1)e^{-R/\tau}} + \frac{e^{-cb}}{1+(|\mathcal{Y}|-1)e^{-R/\tau}}$$

If $z = y^*$,

$$Q'(z|x) = p(y^*|x)q(z|y^*) + \sum_{y\ne y^*} p(y|x)q(z|y) \ge p(y^*|x)q(z|y^*)$$

From Lemma 3 and Lemma 4,

$$Q'(z|x) \ge \frac{1-e^{-cb}}{1+(|\mathcal{Y}|-1)e^{-\gamma R/\tau}}$$

**Lemma 6.**

$$\begin{array}{ccccl}
0 & \le & \mathrm{E}[r(z,Y)/\tau] & \le & P(y^*|x)r(z,y^*)/\tau + e^{-cb}R/\tau \quad , \text{if } z \ne y^* \\
P(y^*|x)r(y^*,y^*)/\tau & \le & \mathrm{E}[r(z,Y)/\tau] & \le & R/\tau \quad , \text{if } z = y^*
\end{array}$$

*Proof.*

$$\mathrm{E}[r(z,Y)/\tau] = \sum_{y\in\mathcal{Y}} P(y|x)r(z,y)/\tau$$

Since for every $y, y' \in \mathcal{Y}, 0 \le r(y,y') \le R$, we have

$$0 \le \mathrm{E}[r(z,Y)/\tau] \le R/\tau$$

If $z \neq y^*$,

$$\mathrm{E}[r(z,Y)/\tau] = P(y^*|x)r(z,y^*)/\tau + \sum_{y \neq y^*} P(y|x)r(z,y)/\tau \leq P(y^*|x)r(z,y^*)/\tau + e^{-cb}R/\tau$$

If $z = y^*$,

$$\mathrm{E}[r(z,Y)/\tau] = P(y^*|x)r(z,y^*)/\tau + \sum_{y \neq y^*} P(y|x)r(z,y)/\tau \geq P(y^*|x)r(y^*,y^*)/\tau$$

□

**Lemma 7.**

$$\begin{array}{ccccc}
\frac{1}{1+(|\mathcal{Y}|-1)e^{R/\tau}} & \leq & Q(z|x) & \leq & \frac{1}{e^{\alpha R/\tau}+(|\mathcal{Y}|-1)e^{-R/\tau}} & , \text{if } z \neq y^* \\
\frac{1}{1+(|\mathcal{Y}|-1)e^{-\alpha R/\tau}} & \leq & Q(z|x) & \leq & \frac{1}{1+(|\mathcal{Y}|-1)e^{-R/\tau}} & , \text{if } y = y^*
\end{array}$$

*where $\alpha = \gamma - (1+\gamma)e^{-cb} < \gamma$.*

*Proof.*

$$Q(z|x) = \frac{e^{\mathrm{E}[r(z,Y)/\tau]}}{\sum_{y' \in \mathcal{Y}} e^{\mathrm{E}[r(y',Y)/\tau]}}$$

From Lemma 6,

$$\frac{1}{1+(|\mathcal{Y}|-1)e^{R/\tau}} \leq Q(z|x) \leq \frac{1}{1+(|\mathcal{Y}|-1)e^{-R/\tau}}$$

If $z \neq y^*$,

$$\begin{aligned}
Q(z|x) &\leq \frac{e^{\mathrm{E}[r(z,Y)/\tau]}}{e^{\mathrm{E}[r(y^*,Y)/\tau]}+|\mathcal{Y}|-1} \leq \left(e^{\mathrm{E}[r(y^*,Y)/\tau]-\mathrm{E}[r(z,Y)/\tau]} + (|\mathcal{Y}|-1)e^{-R/\tau}\right)^{-1} \\
&\leq \left(e^{P(y^*|x)(r(y^*,y^*)/\tau-r(z,y^*)/\tau)} - e^{-cbR/\tau} + (|\mathcal{Y}|-1)e^{-R/\tau}\right)^{-1} \\
&\leq \left(e^{(1-e^{-cb})\gamma R/\tau - e^{-cb}R/\tau} + (|\mathcal{Y}|-1)e^{-R/\tau}\right)^{-1} \\
&= \frac{1}{e^{\alpha R/\tau}+(|\mathcal{Y}|-1)e^{-R/\tau}}
\end{aligned}$$

If $z = y^*$,

$$Q(z|x) = \left(1 + \sum_{y \neq y^*} e^{\mathrm{E}[r(y,Y)/\tau]-\mathrm{E}[r(y^*,Y)/\tau]}\right)^{-1} \leq \frac{1}{1+(|\mathcal{Y}|-1)e^{-\alpha R/\tau}}$$

□

Now, we can prove Theorem 2 with the above lemmas.

**Proof of Theorem 2**

*Proof.*

$$\begin{aligned}
\mathrm{KL}(Q(\cdot|X=x)\|Q'(\cdot|X=x)) &= \sum_{y \in \mathcal{Y}} Q(y|x)\log\frac{Q(y|x)}{Q'(y|x)} = Q(y^*|x)\log\frac{Q(y^*|x)}{Q'(y^*|x)} + \sum_{y \neq y^*} Q(y|x)\log\frac{Q(y|x)}{Q'(y|x)} \\
&\leq \left(1+(|\mathcal{Y}|-1)e^{-R/\tau}\right)^{-1}\log\left(\frac{1}{1-e^{-cb}}\frac{1+(|\mathcal{Y}|-1)e^{-\gamma R/\tau}}{1+(|\mathcal{Y}|-1)e^{-R/\tau}}\right) \\
&\quad + \frac{|\mathcal{Y}|-1}{e^{\alpha R/\tau}+(|\mathcal{Y}|-1)e^{-R/\tau}}\log\frac{1+(|\mathcal{Y}|-1)e^{R/\tau}}{e^{\alpha R/\tau}+(|\mathcal{Y}|-1)e^{-R/\tau}} \\
\lim_{\tau \to 0}\mathrm{KL}(Q(\cdot|X=x)\|Q'(\cdot|X=x)) &\leq \log\frac{1}{1-e^{-cb}} + \lim_{\tau \to 0}\frac{|\mathcal{Y}|-1}{e^{\alpha R/\tau}}\log(|\mathcal{Y}|-1)e^{(1-\alpha)R/\tau} \\
&= \log\frac{1}{1-e^{-cb}} + \lim_{\tau \to 0}\frac{|\mathcal{Y}|-1}{e^{\alpha R/\tau}}\left(\log(|\mathcal{Y}|-1)+(1-\alpha)R/\tau\right) \\
&= \log\frac{1}{1-e^{-cb}} = \log 1 + \frac{e^{-cb}}{1-e^{-cb}} \\
&\leq \frac{e^{-cb}}{1-e^{-cb}} = \frac{1}{1+e^{cb}}
\end{aligned}$$

□

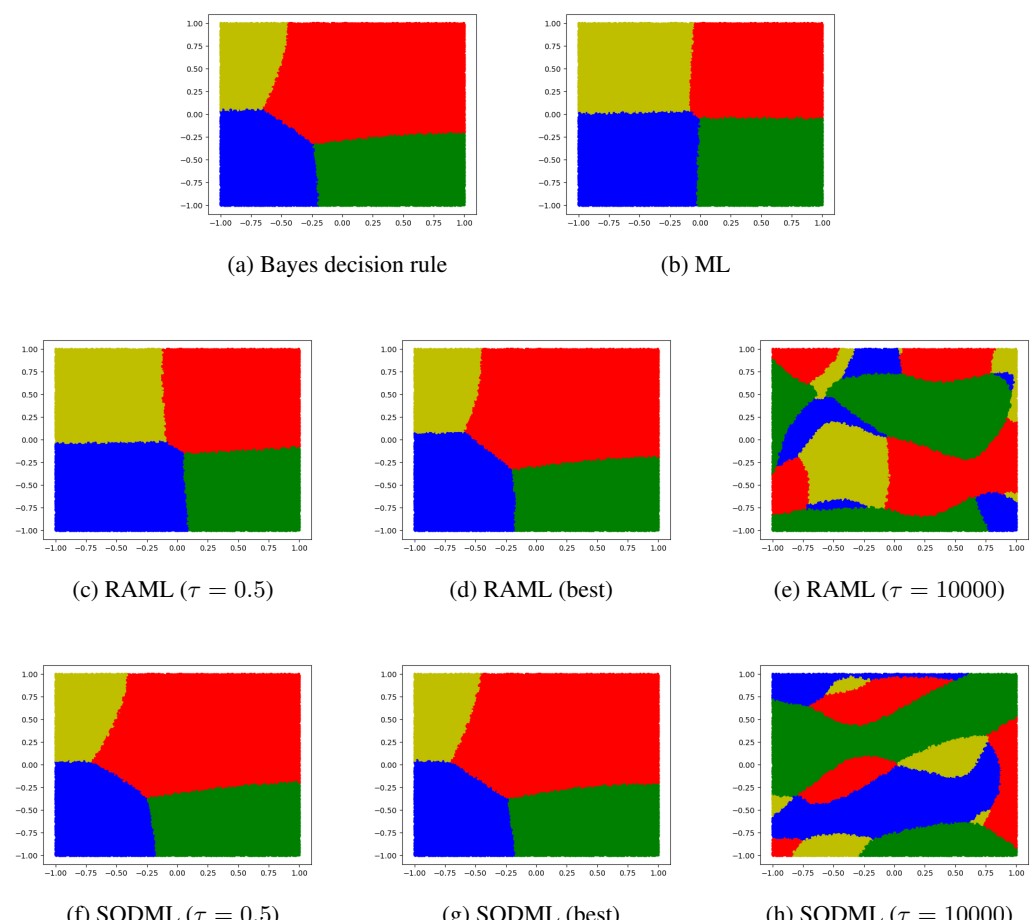

Figure 4: Decision boundaries of different models, together with the Bayes decision rule in (a). (b) display the decision boundary of ML. (c), (d), (e) are the decision boundaries of RAML with $\tau = 0.5$, $\tau = 10000$ and the one achieves the best performance $\tau = 2.4$. (f), (g), (h) are the corresponding boundaries of SQDML. The best performance is achieved with $\tau = 1.1$

## B   COST-SENSITIVE MULTI-CLASS CLASSIFICATION

To better illustrate the properties of ML, RAML and SQDML, we display the decision boundary of the learned models in Figure 4. Figure 4a gives the boundary of the Bayes decision rule, and Figure 4b is the boundary of ML. We can see that, as expected, ML gives "unbiased" boundary because it does not incorporating any information of the task-specific reward.

Figure 4c and 4f are the decision boundaries of RAML and SQDML with $\tau = 0.5$. We can see that, even with small $\tau$, SQDML is able to achieve good decision boundary similar to that of the Bayes decision rule, while the boundary of RAML is similar to that of ML. This might suggest that RAML, as an approximation of SQDML, might "degenerates" to ML due to approximation error.

Figure 4d and 4g provide the boundary of RAML and SQDML that achieve the best performance ($\tau = 2.4$ for RAML and $\tau = 1.1$ for SQDML). RAML is able to produce surprisingly good decisions with proper $\tau$, which is comparable with SQDML.

Figure 4e and 4h are the decision boundaries of RAML and SQDML with large $\tau = 10000$. We can see that, consistently matching our analysis, neither RAML or SQDML can learn reasonable prediction function. The reason is, as we discussed in §3.5, when $\tau$ becomes larger, the softmax

Q-distribution becomes closer to the uniform distribution, providing less information of prediction, even though the approximation error tends to be zero.

## C  IMAGE CAPTIONING WITH MULTIPLE REFERENCES

**Encoder** Following (Yang et al., 2016), we adopt the widely-used CNN architecture VGGNet (Simonyan & Zisserman, 2015) as the image encoder. Specifically, we use the last fully connected layer fc7 as image representation (4096-dimensional), which is further fed into a decoder to generate captions. We use a pre-trained VGGNet model[4], and keep it fixed during the training of decoder.

**Decoder** We use an LSTM network as the decoder to predicate a sequence of target words: $\{y_1, y_2, \ldots, y_T\}$. Formally, the decoder uses its internal hidden state $\mathbf{s}_t$ at each time step to track the generation process, defined as

$$\mathbf{s}_t = f_{\text{LSTM}}(\mathbf{y}_{t-1}, \mathbf{s}_{t-1}),$$

where $\mathbf{y}_{t-1}$ is the embedding of the previous word $y_{t-1}$. We initialize the memory cell of the decoder by passing the fixed-length image representation $\mathbf{x}$ through an affine transformation layer. The probability of the target word $y_t$ is then given by

$$p(y_t|y_{<t}, \mathbf{x}) = \text{softmax}(\mathbf{W}_s \mathbf{s}_t).$$

**Training by $N$-gram Replacement** As discussed in §4.1.2, we approximate the exponentially large hypotheses space $\mathcal{Y}$ using a subset of sampled hypotheses $\mathcal{S}$. Formally, the training set $\mathcal{D}$ consists of pairs of images $x$ and multiple references $\{y^*\}$, i.e., $\mathcal{D} = \{\langle x, \{y^*\}\rangle\}$. For RAML, we split a single training instance $\langle x, \{y^*\}\rangle$ into multiple ones by pairing $x$ with each $y^*$, i.e., $\{\langle x, y^*\rangle\}$. And for each instance $\langle x, y^*\rangle$ we maximize

$$\sum_{y \in \mathcal{Y}} q(y|y^*; \tau) \log P_\theta(y|x) = \sum_{y \in \mathcal{Y}} \frac{\exp\left(r(y, y^*)/\tau\right)}{\sum_{y' \in \mathcal{Y}} \exp\left(r(y', y^*)/\tau\right)} \log P_\theta(y|x)$$

$$\approx \sum_{y \in \mathcal{S}} \frac{\exp\left(r(y, y^*)/\tau\right)}{\sum_{y' \in \mathcal{S}} \exp\left(r(y', y^*)/\tau\right)} \log P_\theta(y|x).$$

For SQDML, for each training example $\langle x, \{y^*\}\rangle$ we directly maximize the weighted log-likelihood w.r.t. the softmax-Q distribution

$$\sum_{y \in \mathcal{Y}} Q(y|x; \tau) \log P_\theta(y|x) = \sum_{y \in \mathcal{Y}} \frac{\exp\left(\sum_{y^*} \frac{1}{|\{y^*\}|} r(y, y^*)/\tau\right)}{\sum_{y' \in \mathcal{Y}} \exp\left(\sum_{y^*} \frac{1}{|\{y^*\}|} r(y', y^*)/\tau\right)} \log P_\theta(y|x)$$

$$\approx \sum_{y \in \mathcal{S}} \frac{\exp\left(\sum_{y^*} \frac{1}{|\{y^*\}|} r(y, y^*)/\tau\right)}{\sum_{y' \in \mathcal{S}} \exp\left(\sum_{y^*} \frac{1}{|\{y^*\}|} r(y', y^*)/\tau\right)} \log P_\theta(y|x).$$

In our experiments, the size of the sampled targets $\mathcal{S}$ is 500 for SQDML and 100 for RAML[5]. For the sake of efficiency, at each iteration of stochastic gradient descent, we only use $k$ randomly-selected hypotheses from $\mathcal{S}$ to perform gradient update. $k$ is 50 for SQDML and 10 for RAML.

**Configuration** We use the sentence-level BLEU with NIST geometric smoothing as the reward. We replace word types whose frequency is less than five with a special `<unk>` token. The resulting vocabulary size is 10,102. The dimensionality of word embeddings and LSTM hidden sates is 256 and 512, respectively. For decoding, we use beam search with a beam size of 5. We use a batch size of 10 for the ML baseline and a larger size of 100 for SQDML and RAML for the sake of efficiency.

## D  EXPERIMENTS ON STRUCTURED PREDICTION

### D.1  DATASET STATISTICS

We present statistics of the datasets we used in Table 7.

---

[4]Downloaded from `https://github.com/kimiyoung/review_net`
[5]Since each image has around five reference captions, this ensures that the number of sampled candidate $y$'s for each training example is roughly the same for SQDML and RAML.

| Dataset | | CoNLL2003 | PTB | IWSLT2014 |
|---|---|---|---|---|
| TRAIN | #Sent | 14,987 | 39,832 | 153,326 |
| | #Token | 204,567 | 843,029 | 2,687,420 / 2,836,554 |
| DEV. | #Sent | 3,466 | 1,700 | 6,969 |
| | #Token | 51,578 | 35,508 | 122,327 / 129,091 |
| TEST | #Sent | 3,684 | 2,416 | 6,750 |
| | #Token | 46,666 | 49,892 | 125,738 / 131,141 |

Table 7: Dataset statistics. #Sent and #Token refer to the number of sentences and tokens in each data set, respectively (for IWSLT, they refer to the number of sentence pairs and tokens of source/target languages).

## D.2 MODELS FOR STRUCTURED PREDICTION

### D.2.1 LOG-LINEAR MODEL

A commonly used log-linear model defines a family of conditional probability $P_\theta(y|x)$ over $\mathcal{Y}$ with the following form:

$$P_\theta(y|x) = \frac{\Phi(y, x; \theta)}{\sum\limits_{y' \in \mathcal{Y}} \Phi(y', x; \theta)} = \frac{\exp(\theta^T \phi(y, x))}{\sum\limits_{y' \in \mathcal{Y}} \exp(\theta^T \phi(y', x))} \quad (4)$$

where $\phi(y, x)$ are the feature functions, $\theta$ are parameters of the model and $\Phi(y, x; \theta)$ captures the dependency between the input and output variables. We define the *partition function*: $Z(x; \theta) = \sum\limits_{y' \in \mathcal{Y}} \exp(\theta^T \phi(y', x))$. Then, the conditional probability in (4) can be written as:

$$P_\theta(y|x) = \frac{\exp(\theta^T \phi(y, x))}{Z(x; \theta)}$$

Now, the objective of RAML for one training instance $(x, y)$ is:

$$\mathcal{L}(\theta) = -\sum_{y' \in \mathcal{Y}} q(y'|y; \tau) \log P_\theta(y'|x) = -\theta^T \left\{ \sum_{y' \in \mathcal{Y}} q(y'|y; \tau) \phi(y', x) \right\} + \log Z(x; \theta) \quad (5)$$

and the gradient is:

$$\begin{aligned} \frac{\partial \mathcal{L}(\theta)}{\partial \theta} &= -\sum_{y' \in \mathcal{Y}} q(y'|y; \tau) \phi(y', x) + \frac{\partial \log Z(x; \theta)}{\partial \theta} \\ &= -\sum_{y' \in \mathcal{Y}} q(y'|y; \tau) \phi(y', x) + \sum_{y' \in \mathcal{Y}} P_\theta(y'|x) \phi(y', x) \\ &= \sum_{y' \in \mathcal{Y}} \left( P_\theta(y'|x) - q(y'|y; \tau) \right) \phi(y', x) \end{aligned} \quad (6)$$

To optimize $\mathcal{L}(\theta)$, we need to efficiently compute the objective and its gradient. In the next two sections, we see that when the feature $\phi(y, x)$ and the reward $r(y, y^*)$ follow some certain factorizations, efficient dynamic programming algorithms exist.

### D.2.2 SEQUENCE CRF

In sequence CRF, $\Phi$ usually factorizes as sum of *potential functions* defined on pairs of successive labels:

$$\Phi(y, x; \theta) = \prod_{i=1}^{L} \psi_i(y_{i-1}, y_i, x; \theta)$$

where $\psi_i(y_{i-1}, y_i, x; \theta) = \exp(\theta^T \phi_i(y_{i-1}, y_i, x))$. When we use the token level label accuracy as reward, the reward function can be factorized as:

$$r(y, y^*) = \sum_{i=1}^{L} \mathbb{I}(y_i = y_i^*)$$

where $y_i$ is the label of the $i$th token (word). Then, the objective and gradient in (5) and (6) can be computed by using the forward-backward algorithm (Wallach, 2004).

### D.2.3 EDGE-FACTORIZED TREE-STRUCTURE MODEL

In dependency parsing, $y$ represents a generic dependency tree which consists of directed edges between heads and their dependents (modifiers). The edge-factorized model factorizes potential function $\Phi$ into the set of edges:

$$\Phi(y, x; \theta) = \prod_{e \in y} \psi_e(e, x; \theta)$$

where $e$ is an edge belonging to the tree $y$. $\psi_e(e; \theta) = \exp(\theta^T \phi_e(e, x))$. The reward of UAS can be factorized as:

$$r(y, y^*) = \sum_{i=1}^{L} \mathbb{I}(y_i = y_i^*)$$

where $y_i$ is the head of the $i$th word in the sentence $x$. Then, we have:

$$
\begin{aligned}
\sum_{y \in \mathcal{Y}} P_\theta(y|x) \phi(y, x) &= \sum_{y \in \mathcal{Y}} \sum_{e \in y} P_\theta(y|x) \phi_e(e, x) \\
&= \sum_{e \in \mathcal{E}} \phi_e(e, x) \left\{ \sum_{y \in \mathcal{Y}(e)} P_\theta(y|x) \right\}
\end{aligned}
\tag{7}
$$

where $\mathcal{E}$ is the set of all possible edges for sentence $x$ and $\mathcal{Y}(e) = \{y \in \mathcal{Y} : e \in y\}$. With similar derivation, we have

$$\sum_{y' \in \mathcal{Y}} q(y'|y; \tau) \phi(y', x) = \sum_{e \in \mathcal{E}} \phi_e(e, x) \left\{ \sum_{y' \in \mathcal{Y}(e)} q(y'|y) \right\} \tag{8}$$

Both (7) and (8) can be computed by using the inside-outside algorithm (Paskin, 2001; Ma & Zhao, 2012)

### D.2.4 ATTENTIONAL NEURAL MACHINE TRANSLATION MODEL

**Model Overview**

We apply a neural encoder-decoder model with attention and input feeding (Luong et al., 2015). Given a source sentence $x$ of $N$ words $\{x_i\}_{i=1}^{N}$, the conditional probability of the target sentence $y = \{y_i\}_{i=1}^{T}$, $p(y|x)$, is factorized as $p(y|x) = \prod_{t=1}^{T} p(y_t|y_{<t}, x)$. The probability is computed using a bi-directional LSTM encoder and an LSTM decoder:

**Encoder** Let $\mathbf{x}_i$ denote the embedding of the $i$-th source word $x_i$. We use two unidirectional LSTMs to process $x$ in forward and backward order, and get the sequence of hidden states $\{\vec{\mathbf{h}}_i\}_{i=1}^{N}$ and $\{\overleftarrow{\mathbf{h}}_i\}_{i=1}^{N}$ in the two directions:

$$
\begin{aligned}
\vec{\mathbf{h}}_i &= f_{\overrightarrow{\text{LSTM}}}(\mathbf{x}_i, \vec{\mathbf{h}}_{i-1}) \\
\overleftarrow{\mathbf{h}}_i &= f_{\overleftarrow{\text{LSTM}}}(\mathbf{x}_i, \overleftarrow{\mathbf{h}}_{i+1}),
\end{aligned}
$$

where $f_{\overrightarrow{\text{LSTM}}}$ and $f_{\overleftarrow{\text{LSTM}}}$ are standard LSTM update functions as in Hochreiter & Schmidhuber (1997). The representation of the $i$-th word, $\mathbf{h}_i$, is then given by concatenating $\vec{\mathbf{h}}_i$ and $\overleftarrow{\mathbf{h}}_i$.

**Decoder** An LSTM is used as the decoder to predict a target word $y_t$ at each time step $t$. Formally, the decoder maintains a hidden state $\mathbf{s}_t$ to track the translation process, defined as

$$\mathbf{s}_t = f_{\text{LSTM}}([\mathbf{y}_{t-1} : \tilde{\mathbf{s}}_{t-1}], \mathbf{s}_{t-1}),$$

where $[:]$ denotes vector concatenation, and $\mathbf{y}_{t-1}$ is the embedding of the previous target word. We initialize the first memory cell of the decoder using the last hidden states of the two encoding LSTMs: $\mathbf{cell}_0 = \mathbf{W}[\vec{\mathbf{h}}_N : \overleftarrow{\mathbf{h}}_N] + \mathbf{b}$. And the first hidden state of the decoder is initialized as $\mathbf{s}_0 = \tanh(\mathbf{cell}_0)$. The attentional vector $\tilde{\mathbf{s}}_t$ is computed as

$$\tilde{\mathbf{s}}_t = \tanh(\mathbf{W}_c[\mathbf{s}_t : \mathbf{c}_t]),$$

| Method | BLEU |
|---|---|
| ML Baseline | 27.66 |
| $\tau = 0.10$ | 28.22 |
| $\tau = 0.20$ | 28.22 |
| $\tau = 0.30$ | 28.22 |
| $\tau = 0.40$ | 28.31 |
| $\tau = 0.50$ | 28.28 |
| $\tau = 0.60$ | 28.23 |
| $\tau = 0.70$ | **28.61** |
| $\tau = 0.80$ | 28.30 |
| $\tau = 0.90$ | 28.40 |
| $\tau = 1.00$ | 28.42 |
| $N$-GRAM | **28.77** |

Table 8: Corpus-level BLEU score of RAML using importance sampling

| Method | BLEU |
|---|---|
| ML Baseline | 27.66 |
| $\tau = 0.60$ | 27.96 |
| $\tau = 0.65$ | 27.94 |
| $\tau = 0.70$ | 28.18 |
| $\tau = 0.75$ | 27.96 |
| $\tau = 0.80$ | 27.93 |
| $\tau = 0.85$ | 27.97 |
| $\tau = 0.90$ | **28.39** |
| $\tau = 0.95$ | 28.30 |
| $\tau = 1.00$ | 28.32 |
| $\tau = 1.05$ | 27.92 |
| IMPT. SAMPLE | 28.61 |
| $N$-GRAM | **28.77** |

Table 9: Corpus-level BLEU score of RAML using negative Hamming distance as the reward function

where the context vector $\mathbf{c}_t$ is a weighted sum of the source encodings $\{\mathbf{h}_i\}$ via attention (Bahdanau et al., 2015). The probability of the target word $y_t$ is then given by

$$p(y_t|y_{<t}, x) = \text{softmax}(\mathbf{W}_s \tilde{\mathbf{s}}_t).$$

**Configuration** We use the same pre-processed dataset as in Wiseman & Rush (2016). The vocabulary size of the German and English data is 32,008 and 22,821 words, resp. Similar as Bahdanau et al. (2017), the dimensionality of word embeddings and LSTM hidden states is 256. All neural network parameters are uniformly initialized between $[-0.1, +0.1]$. We use Adam optimizer. We validate the perplexity of the development set after every epoch, and halve the learning rate if the validation performance drops. We use the sentence level BLEU with NIST geometric smoothing as the training reward, and use the official `multi-bleu.perl` script for evaluating corpus-level BLEU. The beam size for decoding is 5. We use a batch size of 64 for ML baseline and a larger size of 100 for RAML for the sake of efficiency.

**Approximating the Learning Objective using Importance Sampling**

As suggested in Norouzi et al. (2016), with BLEU as the training reward, the objective function (4) of RAML could be optimized using importance sampling. To verify this, we conducted experiment using importance sampling. Since we cannot directly sample from the exponentiated payoff distribution parameterized by BLEU score (i.e., $q_{\text{BLEU}}(y|y^*, \tau)$), we use the payoff distribution with negative Hamming distance (i.e., $q_{\text{hm}}(y|y^*, \tau)$) as the proposal distribution, and sample from $q_{\text{hm}}(y|y^*, \tau)$

instead. Specifically, at each training iteration, we approximate (4) by

$$
\sum_{y \in \mathcal{Y}} q(y|y^*;\tau) \log P_\theta(y|x_i) \approx \sum_{y \sim q_{\text{hm}}(\cdot|y^*,\tau)} \frac{\tilde{q}_{\text{BLEU}}(y|y^*;\tau)/\tilde{q}_{\text{hm}}(y|y^*;\tau)}{\sum_{y' \sim q_{\text{hm}}(\cdot|y^*,\tau)} \tilde{q}_{\text{BLEU}}(y'|y^*;\tau)/\tilde{q}_{\text{hm}}(y'|y^*;\tau)} \log P_\theta(y|x_i)
$$

$$
= \sum_{y \sim q_{\text{hm}}} \frac{\exp\{\text{BLEU}(y,y^*)/\tau\}/\exp\{\text{hm}(y,y^*)/\tau\}}{\sum_{y' \sim q_{\text{hm}}} \exp\{\text{BLEU}(y',y^*)/\tau\}/\exp\{\text{hm}(y',y^*)/\tau\}} \log P_\theta(y|x_i),
$$

where the $\tilde{q}(\cdot)$'s denote the payoff distributions without normalization terms. $\text{BLEU}(\cdot)$ and $\text{hm}(\cdot)$ denote sentence-level BLEU score and negative Hamming distance, respectively. We use a sample size of 10.[6]

To draw a sample $y$ from $q_{\text{hm}}(y|y^*,\tau)$, we follow Norouzi et al. (2016) and apply stratified sampling. We first sample a distance $d \in [0, 1, 2, \ldots, |y^*|-1, |y^*|]$, and then sample a sentence $y$ with Hamming distance $d$ from $y^*$. Let $c(d, L)$ denote the number of $y$'s with length $L$ and an Hamming distance of $d$ from the ground-truth $y^*$, $q_{\text{hm}}(y|y^*,\tau)$ is then defined as:

$$
q_{\text{hm}}(y|y^*,\tau) = \frac{\exp\{\text{hm}(y,y^*)/\tau\}}{\sum_{d=0}^{|y^*|} c(d,|y^*|) \cdot \exp\{-d/\tau\}}.
$$

Similar as in Norouzi et al. (2016), $c(d, L)$ is approximated by considering $d$ substitutions of words from $y^*$:

$$
c(d, L) = \binom{L}{d}(V-1)^d,
$$

where $V$ is the vocabulary size.[7]

Table 8 lists the performance of importance sampling with different temperatures. The best model (under $\tau = 0.8$) is comparable with the one achieved by $n$-gram replacement. However, $n$-gram replacement is much simpler to implement, and importance sampling requires extra computation of the proposal distribution and associated importance weights, which would be less computationally efficient. In our experiments, our highly optimized RAML model achieves a training speed of 18,000 words/sec for importance sampling and 21,000 words/sec for $n$-gram replacement.

**Extra Experiments using Negative Hamming Distance as Training Reward**

For the sake of completeness, we also experimented using the negative Hamming distance as the reward function for RAML, as proposed in Norouzi et al. (2016). Results are listed in Table 9. The best model gets a corpus-level BLEU score of 28.39, which is worse than the best results achieved by optimizing directly towards BLEU scores (c.f. Table 4).

---

[6]We also tried larger sample size but did not observe significant gains.

[7]Through correspondence with the authors, we scale $\tau$ by $\frac{1}{1+\log(V-1)}$

