# OpenReview forum: "Softmax Q-Distribution Estimation for Structured Prediction: A Theoretical Interpretation for RAML"
_ICLR.cc/2018/Conference — Reject_

### Official Review · AnonReviewer1 · 2017-11-23

**Rating:** 5
**Confidence:** 2

**Review:**

This paper dives deeper into understand reward augmented maximum likelihood training. Overall, I feel that the paper is hard to understand and that it would benefit from more clarity, e.g., section 3.3 states that decoding from the softmax q-distribution is similar to the Bayes decision rule. Please elaborate on this.

Did you compare to minimum bayes risk decoding which chooses the output with the lowest expected risk amongst a set of candidates?

Section 4.2.2 says that Ranzato et al. and Bahdanau et al. require sampling from the model distribution. However, the methods analyzed in this paper also require sampling (cf. Appendix D.2.4 where you mention a sample size of 10). Please explain the difference.

---

> ### Author Response · Authors · 2017-12-10
> **Response to review**
>
> We thank for your time and insightful comments.
>
> In section 3.3 we claim that decoding from the softmax Q-distribution delivers the Bayes decision rule, because via decoding from the softmax Q-distribution in Eq (12), we directly get the prediction function h(x) = \argmax_{y \in Y} Q(y|X=x; \tau) = \argmax_{y \in Y} E_{P(Y|X=x)}[r(y, Y)], which is just the Bayes decision rule. Eq (13) gives the details of the derivation.
>
> We discussed the relation and difference between minimum risk decoding and our method. Both RAML and SQDML are trying to learn distributions, decoding from which (approximately) provide the Bayes decision rule, while minimum risk decoding, on the other hand, attempts to (approximately) estimate the Bayes decision rule directly by compution the expectation w.r.t the learned distribution.
> For experiments, we did not compare our methods with it.
>
> The main difference of sampling between SQDML (or RAML) and RL-based approaches (Ranzato et al, 2016, Bahdanau et al, 2017) is that RL-based approaches require to sample from the learned model distribution which keeps updating during the training procedure. Thus, these approaches suffers from high variance, and usually require a pre-trained ML baseline to initialize model. SQDML (or RAML), however, does sampling from a fixed distribution (the pay-off distribution for RAML or the empirical softmax Q-distribution for SQDML), making the training procedure more stable and requring no pre-training initialization.

---

### Official Review · AnonReviewer3 · 2017-11-27
**some interesting ideas, but the paper need to be strengthened**

**Rating:** 5
**Confidence:** 4

**Review:**

This paper interprets reward augmented maximum likelihood followed by decoding with the most likely output as an approximation to the Bayes decision rule.

I have a few questions on the motivation and the results.
- In the section "Open Problems in RAML", both (i) and (ii) are based on the statement that the globally optimal solution of RAML is the exponential payoff distribution q. This is not true. The globally optimal solution is related to both the underlying data distribution P and q, and not the same as q. It is given by q'(y | x, \tau) = \sum_{y'} P(y' | x) q(y | y', \tau).
- Both Theorem 1 and Theorem 2 do not directly justify that RAML has similar reward as the Bayes decision rule. Can anything be said about this? Are the KL divergence small enough to guarantee similar predictive rewards?
- In Theorem 2, when does the exponential tail bound assumption hold?
- In Table 1, the differences between RAML and SQDML do not seem to support the claim that SQDML is better than RAML. Are the differences actually significant? Are the differences between SQDML/RAML and ML significant? In addition, how should \tau be chosen in these experiments?

---

> ### Author Response · Authors · 2017-12-10
> **Response to review**
>
> We thank for your time and insightful comments.
>
> For your first quesiton about our statment of the globally optimal solution of RAML, we concede that the globally optimal solution of RAML is P_{RAML}{y|X=x) = E_{P(Y|X=x)}[q(y|Y)], which is just the Q' distribution in Eq (16) in section 3.4.
> To post the questions in the paragraph "Open Problems in RAML", we want to point out that from the original form of RAML in Eq (4) and (6), it is not straight-forward to understand the behavior of the pay-off q distribution in Eq (5), neither the globally optimal solution of RAML. Moreover, as pointed out in quesiton (iii), there is no rigorous theorectical evidence showing that RAML provides a better prediction function. From our Softmax Q-distribution estimation framework in section 3, we derived that the globally optimal solution of RAML is actually Q' in Eq (16), and linked the Q' distribution with our softmax Q distribution in Eq (12) by providing two KL-based bounds in Theorem 1 and 2, demonstrating that Q' is approximating to Q. We really appreciate your comment about the confusion of this part and will definitely revise this paragraph to make the problems of RAML more clear.
>
> For your question about Theorem 1 and 2, both of them are trying to characterizing the approximating error from Q' to Q by upper-bounding the KL divergence between them. Since the Q and Q' distributions are the "target" distributions that SQDML and RAML are learning, respectively, and the prediction functions of SQDML and RAML are directly generated by decoding from these two distributions, we can say that SQDML and RAML should have similar predictive rewards if Q and Q' are close. And decoding from Q distribution delivers Bayes decision rule, thus Theorem 1 and 2 guarantee that RAML would have similar predictive rewards when the assumptions hold. Further, our experiment on syntactic data empirically demonstrate that RAML is able to achieve similar predictive rewards with SQDML which asymptotically achieves Bayes decision rule.
>
> The exponential tail bound assumption holds when the conditional distribution P(Y|X=x) is close to a deterministic distribution w.r.t a 'ground-truth' y*.
>
> Results in Table 1 illustrate that SQDML achieves better performance towards the oprimized metric. We conceded in the paper that the improvements of SQDML over RAML are not significant, and gave a possible explanation that the reference captions for each image are largely different, making it highly non-trivial for the model to predict a "consensus" caption that agrees with multiple references. Examples are given in Figure 3.
>
> The selection of \tau in practice depends on the task and the properties of reward funciton r(y, y*). In our experiments on different NLP structured prediction tasks, we found that choosing \tau in the region of (0.5, 1.5) results pretty good performance. One may perform fine-tuning of \tau on the validation sets.

---

### Official Review · AnonReviewer2 · 2017-12-01
**good theoretical interpretation of RAML, weak experiment results**

**Rating:** 6
**Confidence:** 3

**Review:**

The authors claim three contributions in this paper. (1) They introduce the framework of softmax Q-distribution estimation, through which they are able to interpret the role the payoff distribution plays in RAML. Specifically, the softmax Q-distribution serves as a smooth approximation to the Bayes decision boundary. The RAML approximately estimates the softmax Q-distribution, and thus approximates the Bayes decision rule. (2) Algorithmically, they further propose softmax Q-distribution maximum likelihood (SQDML) which improves RAML by achieving the exact Bayes decision boundary asymptotically. (3) Through one experiment using synthetic data on multi-class classiﬁcation and one using real data on image captioning, they show that SQDML is consistently as good or better than RAML on the task-speciﬁc metrics that is desired to optimize.

I found the first contribution is sound, and it reasonably explains why RAML achieves better performance when measured by a specific metric. Given a reward function, one can define the Bayes decision rule. The softmax Q-distribution (Eqn. 12) is defined to be the softmax approximation of the deterministic Bayes rule. The authors show that the RAML can be explained by moving the expectation out of the nonlinear function and replacing it with empirical expectation (Eqn. 17). Of course, the moving-out is biased but the replacing is unbiased.

The second contribution is partially valid, although I doubt how much improvement one can get from SQDML. The authors define the empirical Q-distribution by replacing the expectation in Eqn. 12 with empirical expectation (Eqn. 15). In fact, this step can result in biased estimation because the replacement is inside the nonlinear function. When x is repeated sufficiently in the data, this bias is small and improvement can be observed, like in the synthetic data example. However, when x is not repeated frequently, both RAML and SQDML are biased. Experiment in section 4.1.2 do not validate significant improvement, either.

The numerical results are relatively weak. The synthetic experiment verifies the reward-maximizing property of RAML and SQDML. However, from Figure 2, we can see that the result is quite sensitive to the temperature \tau. Is there any guidelines to choose \tau? For experiments in Section 4.2, all of them are to show the effectiveness of RAML, which are not very relevant to this paper. These experiment results show very small improvement compared to the ML baselines (see Table 2,3 and 5).  These results are also lower than the state of the art performance.

A few questions:
(1). The author may want to check whether (8) can be called a Bayes decision rule. This is a direct result from definition of conditional probability. No Bayesian elements, like prior or likelihood appears here.
(2). In the implementation of SQDML, one can sample from (15) without exactly computing the summation in the denominator. Compared with the n-gram replacement used in the paper, which one is better?
(3). The authors may want to write Eqn. 17 in the same conditional form of Eqn. 12 and Eqn. 14. This will make the comparison much more clear.
(4). What is Theorem 2 trying to convey? Although \tau goes to 0, there is still a gap between Q and Q'. This seems to suggest that for small \tau, Q' is not a good approximation of Q. Are the assumptions in Theorem 2 reasonable? There are several typos in the proof of Theorem 2.
(5). In section 4.2.2, the authors write "the rewards we directly optimized in training (token-level accuracy for NER and UAS for dependency parsing) are more stable w.r.t. τ than the evaluation metrics (F1 in NER), illustrating that in practice, choosing a training reward that correlates well with the evaluation metric is important". Could you explain it in more details?

---

> ### Author Response · Authors · 2017-12-10
> **Response to review**
>
> We thank for your detailed comments and the appreciation of our proposed theoretical interpretation of RAML.
>
> Let us first point out that our main contribution in this work is to provide a theoretical interpretation of RAML. Our experiments are designed to verify our theoretical claims. An exhaustive comparison with current state-of-the-art are however outside the scope of this work (due also to the space limit).
>
> For your concerns about the numerical results of our experiments, from Figure 1 we see that SQDML is quite stable for different values of tau in the region from 0.1 to 3.0. The results in Figure 2 fluctuate when \tau is pretty large, which verifies our discussion in section 3.5 that the softmax Q-distribution becomes closer to the uniform distribution when \tau becomes larger, making it less expressive for prediction. In practice, we found that choosing \tau in the region of (0.5, 1.5) results pretty good performance. One may perform fine-tuning of \tau on the validation sets.
>
> The experiments in section 4.2, as discussed in the first paragraph of this section, is to further confirm the empirical success of RAML (and SQDML) over ML. The models of NER and dependency parsing are classical feature-based statistical models, whose performance are not as good as state-of-the-art neural network models. But our experiments on machine translation are based on the state-of-the-art attentional neural sequence-to-sequence model, and we obtained better performance than Bahdanau, et al. (2017) which incorporated actor-critic algorithm with sequence-to-sequence, demonstrating the effectiveness of RAML. Moreover, from the results in Table 6 in this section, we observed that RAML outperforms ML on the directly optimized metrics while ML gets better results under exact match accuracy. This is in line with our theoretical analysis.
>
> For your questions about details:
> (1) Bayes decision rule is a commonly used terminology in statistical decision theory.
> (2) You said that one can sample from (15) without exactly computing the summation in the denominator. We do not fully understand it. Did you say importance sampling? We have experimental results of importance sampling for machine translation, which is a little bit worse than the n-gram replacement used in this paper. Please see Appendix D 2.4 for details.
> (3) We sincerely appreciate your comment for improving the notation and will revise it in the next version.
> (4) Theorem 1 proved that when \tau becomes larger, the approximation error tends to be zero. At the same time, however, the softmax Q-distribution becomes closer to the uniform distribution, providing less information for prediction. Thus, in practice, we cannot choose a large \tau in order to achieve small approximation error. Theorem 2 is trying to say that under some assumptions, small approximation error is also able to be achieved even with small \tau. The exponential tail bound assumption in Theorem 2 holds when the conditional distribution P(Y|X=x) is close to a deterministic distribution w.r.t a 'ground-truth' y*.
> (5) In the experiments of NER, to efficiently compute the objective function, we use token-level accuracy as the reward which is not exactly the same as the evaluation metric F1 score. For dependency parsing, we use the UAS as the reward which is just the official evaluation metric for this task. According to the results in Table 2 and 3, RAML is quite stable on token-level accuracy and UAS w.r.t \tau, but less stable on F1 score.

---

### Author Response · Authors · 2017-12-29
**Update to the paper**

We revised the paragraph of "Open Problems of RAML" to make the problems of RAML more clear.
Specifically, we merged the first and second problem to a single one.
We also added a "discussion" paragraph before introducing the open issues of RAML.

---

### Decision · Program_Chairs · 2018-01-29
**ICLR 2018 Conference Acceptance Decision**

**Decision:**

Reject

**Comment:**

There are some interesting ideas discussed in the paper, but the reviewers expressed difficulty understanding the motivation and the theoretical results. The experiments do not seem convincing in showing that SQDML achieves significant gains. Overall, the the paper needs either stronger and clearer theoretical results, or more convincing experiments for publication at ICLR.